# Detection and Level Evaluation of Antibodies Specific to Environmental Bacteriophage I11mO19 and Related Coliphages in Non-Immunized Human Sera

**DOI:** 10.3390/antibiotics12030586

**Published:** 2023-03-15

**Authors:** Ewa Brzozowska, Tomasz Lipiński, Agnieszka Korzeniowska-Kowal, Karolina Filik, Andrzej Górski, Andrzej Gamian

**Affiliations:** 1Laboratory of Medical Microbiology, Department of Immunology of Infectious Diseases, Hirszfeld Institute of Immunology and Experimental Therapy, Polish Academy of Sciences, Weigla 12, 53-114 Wrocław, Poland; 2Bioengineering Research Group, Łukasiewicz Research Network-PORT Polish Center for Technology Development, 54-066 Wroclaw, Poland; 3Bacteriophage Laboratory, Hirszfeld Institute of Immunology and Experimental Therapy, Polish Academy of Sciences, Weigla 12, 53-114 Wrocław, Poland

**Keywords:** bacteriophages, anti-phage antibodies, coliphages, cross-reactivity

## Abstract

Bacteriophages (phages) are viruses infecting bacteria. They are widely present in the environment, food, and normal microflora. The human microbiome is a mutually interdependent network of bacteria, bacteriophages, and human cells. The stability of these tri-kingdom interactions may be essential for maintaining immunologic and metabolic health. Phages, as with each other’s antigens, may evoke an immune response during a human’s lifetime and induce specific antibody generation. In this manuscript, we labeled these antibodies as naturally generated. Naturally generated antibodies may be one of the most important factors limiting the efficacy of phage therapy. Herein, we attempted to determine the physiological level of these antibodies specific to a population bacteriophage named I11mO19 in human sera, using an ELISA-based assay. First, we purified the phage particles and assessed the immunoreactivity of phage proteins. Then, affinity chromatography was performed on columns with immobilized phage proteins to obtain a fraction of human polyclonal anti-phage antibodies. These antibodies were used as a reference to elaborate an immunoenzymatic test that was used to determine the level of natural anti-phage antibodies. We estimated the average level of anti-I11mO19 phage antibodies at 190 µg per one milliliter of human serum. However, immunoblotting revealed that cross-reactivity occurs between some proteins of I11mO19 and two other coliphages: T4 and ΦK1E. The antigens probably share common epitopes, suggesting that the determined level of anti-I11mO19 phage might be overestimated and reflects a group of antibodies reactive to a broad range of other *E. coli* phages. Anti-I11mO19 antibodies did not react with *Pseudomonas* bacteriophage F8, confirming specificity to the coliphage group. In this work, we wanted to show whether it is possible to determine the presence and level of anti-phage antibodies in nontargeted-immunized sera, using an immunoenzymatic assay. The conclusion is that it is possible, and specific antibodies can be determined. However, the specificity refers to a broader coliphage group of phages, not only the single phage strain.

## 1. Introduction

Bacteriophages are the most numerous form of life on Earth, ten times more numerous than bacteria. They exist in all environments where bacteria grow: ground and surface water; soil; food; sewage; and sludge. They belong to a microbiome network in humans and animals and are present in feces, urine, saliva, spit, rumen, and serum [1]. They are responsible for 10–80% of total bacterial mortality in aquatic ecosystems and are essential in limiting bacterial populations [2,3]. Phages may contribute to the mechanisms of the organism’s natural defense against harmful bacteria. Our understanding of bacteriophages’ interrelationship (cross-talk) and the human organism is still insufficient. However, the mechanisms by which some phages stimulate antibody production have been described and presented [4]. Anti-phage antibodies are predominately IgM isotypes, but IgG and IgA responses are also induced [5]. We are sure many individuals have preexisting, neutralizing antibodies against different phages, including those commonly used in phage therapy [6]. These antibodies could be relevant to infections, influence immune system regulation, and affect phage therapy. Suitable methods to detect the naturally produced antibodies may be immunoassays such as Western blotting (WB) or ELISA [7]. While using WB, we can determine the presence of specific antibodies. In ELISA, we can evaluate the level of the antibodies [8].

The presented paper focuses on the detection and level determination of anti-phage antibodies to environmental bacteriophage I11mO19. This phage infects pathogenic *E. coli* O19 (PCM 2674) in the human microflora. The goal was to assess whether the phage proteins are immunoreactive and determine the antibody level in nontargeted-immunized human sera. Interestingly, we showed cross-reactivity of the specific anti-I11mO19 antibodies with two other *E. coli* phages: T4 and ΦK1E, while these antibodies were not specific to *Pseudomonas* phage F8.

## 2. Materials and Methods

### 2.1. Bacterial Strains and Bacteriophages

Strains of *Escherichia coli* B (PCM 1935), *E. coli* O19:H-:K- (PCM 2674), *E. coli* O24:K1 (PCM 195), and *Pseudomonas aeruginosa* NBRC 13743 (PCM 2720) were used as hosts for the propagation of the phages T4, I11mO19, ΦK1E, and F8, respectively. We obtained the bacteria from the Polish Collection of Microorganisms (PCM). The bacteriophages were from the Laboratory of Bacteriophages at the Hirszfeld Institute of Immunology and Experimental Therapy of the Polish Academy of Sciences in Wroclaw. I11mO19, as a hospital isolate, was selected for the study’s phages, T4 phage with a receptor rough type LPS receptor, and phage FK1 specific for *E. coli* K1, a human pathogen, with capsular colominic acid as a receptor.

### 2.2. Bacteriophages Propagation and Purification

Bacteriophages were cultured and purified according to Lipiński et al. [9]. The bacterial culture was grown at 37 °C, shaken for 2 to 3 h, and was controlled by measuring the optical density (OD) at λ = 600 nm towards Luria–Bertani Broth (LB) (Sigma-Aldrich L3022, Steinheim, Germany) as a reference. When the OD reached 1.00, the titer of bacterial cells was estimated as 10^8^–10^9^ CFU/mL, and the bacterial culture was centrifuged (5000× *g*, 15 min, 4 °C, Heraeus) in sterile tubes (Sarstedt, 62.547.004, 50 mL). Bacterial cells were suspended in 20 mL of LB and transferred to a bottle (maximal volume of 2000 mL) containing 1200 mL of LB or, in the case of ΦK1E phage, casein hydrolysate broth, Sigma-Aldrich C8845, Steinheim, Germany. Then, we added 1.2 mL of 2.5 M MgCl_2_ (Pol-Aura, 7786-30-3, Olsztyn, Poland) (to the final concentration of 2.5 mM) and 12 mL of 50% glucose (Pol-Aura, 50-99-7, Olsztyn, Poland) (to the final concentration of 0.5%) to the bottle with an aeration system. We incubated the bacterial culture until the OD at λ = 600 nm was 1.00. The next step was bacteria cell infection by phages (the ratio of bacteriophage particles to bacterial cells was 1:1). Five minutes later, we added glucose to the final concentration of 0.5%. In the case of bacteriophage T4 propagation, we added indol (Pol-Aura, 120-72-9, Olsztyn, Poland) to the final concentration of 1 mM. After bacteria lysis by phages (about 4 h), we used chloroform to a final concentration of 1% (*v/v*). The phage culture was cooled and centrifuged (5000× *g*, 5 °C, 15 min). The supernatant was filtered through a filter (Steritop Millipore, Merck, Darmstadt, Germany) with a cut-off of 0.22 µm and stored at 4 °C.

The phage particle purification from the bacterial lysate was performed in a continuous, circulating filtration process, with selective membranes installed on a mini-holder Pellicon device (T186233 Millipore Pellicon-2 Mini Cassette Holder XX42PMINI, Millipore, Taufkirchen, Germany). The membranes used in phage particle purification were PLCXK (Millipore) with a cut-off of 1000 kDa (P2B01MC05) and 100 kDa (PLCHK, P2C100C01, Millipore, Taufkirchen, Germany). The next step was phage particle washing, using 0.05 M Tris/HCl (Sigma, 252859) buffer pH 8.0 containing 0.25 M of NaCl (POCH, 794121422, Polish Chemical Reagents, Gliwice, Poland) and 0.005 M of MgCl_2_ (Pol-Aura, PA-06-612050426, Olsztyn, Poland), then 0.05 M phosphate buffer (pH. 70) (IITD, PAS) containing 5 mM EDTA (or 0.25% sodium deoxycholate (DOC)) (Pol-Aura, PA-03-4011-E#100G, Olsztyn, Poland). Lysozyme (Merk, 9001-63-2, Darmstadt, Germany) was added (38,500 U/mg, Sigma) to the final concentration of 0.05% (*w/v*). Peptidoglycan was digested at 4 °C overnight, and the lysozyme was removed by filtration through a 100 kDa membrane on the Pellicon mini-holder. We monitored the progress of filtration by measuring absorption (A) at λ = 280 nm, and then the specimen was concentrated by ultrafiltration on an Amicon (Merk, C325, Millipore, Taufkirchen, Germany) under nitrogen pressure (membrane 100 kDa PLHK, Millipore). The remnants of lipopolysaccharide (LPS) were removed, using affinity chromatography on High Capacity Endotoxin Removal Resin (Thermo Scientific, 88270, Waltham, MA, USA). The phage concentration was estimated by double-layer methods and was expressed in PFU/mL (Plaque Forming Unit per mL) [10].

### 2.3. Determination of the LPS Content

The endotoxin level was estimated with an endpoint chromogenic Limulus amebocyte lysate assay (LAL) QCL-1000™ (Lonza, 50-648U, Basel, Switzerland). We used a pyrogen-free microplate (Costar, Corning, 3596, Tewksbury, MA, USA) and performed the assay, according to the manufacturer’s recommendations [11]. For each bacteriophage sample, three repeats were made, with three repetitions of spike samples. We included pyrogen-free water samples as negative controls in each analysis. Standard curves were generated with a control standard *E. coli* O111:B4 endotoxin (E50-640) and a LAL reagent, and the curve generated a correlation greater than 0.990. The color reaction was measured at λ = 405 nm with a microplate reader, BIOTEK Power Wale XS, Microplate Spectrophotometer.

### 2.4. Human Sera

Human IgG antibodies were evaluated in sera of 150 healthy volunteers (aged from 18 to 50) who had never been treated with phages. Each volunteer was a blood donor in the Blood Donation Center of the Military Hospital in Wroclaw and signed the agreement for testing. The study protocol was approved by the Medical Ethics Committee of Wroclaw Medical University, Wroclaw, Poland, and the study was conducted in accordance with the Helsinki Declaration of 1975, as revised in 1983. We separated the serum from the blood by clotting (3 h at 37 °C and 4 °C) and centrifugation at 2516× *g* for 20 min at 4 °C. Serum was aliquoted into 100 µL and stored at −20 °C.

### 2.5. Immunoblotting

Immunoblotting (Western blotting) was carried out after sodium dodecyl sulfate electrophoresis (SDS-PAGE) [12]. Electrophoresis of proteins was performed in 12.5% (*w/v*) polyacrylamide gel under denaturing conditions [13]. We used Immobilon-P (PVDF) membrane (Millipore, IPVH00010, Taufkirchen, Germany) in the test. We performed the blocking step, using 1% Bovine Serum Albumin (BSA) (Sigma-Aldrich, 126615, Steinheim, Germany) in TBS-T buffer (Tris Buffered Saline (Sigma, 252859, Steinheim, Germany), with 0.05% (*v/v*) Tween 20 (Sigma-Aldrich, 9005-64-5, Steinheim, Germany) at 37 °C for one hour. Then, the membrane was washed with TBS-T once for 15 min and twice for 5 min. Next, the reaction with human sera (diluted 50–500 times) was performed (30 min at 37 °C and 4 °C overnight in TBS-T with 1% BSA). The antibody excess was removed by washing three times with TBS-T. Then, we incubated the membrane with goat anti-human IgG antibodies conjugated with alkaline phosphatase, diluted to 1:5000 (Sigma, diluted in TBS-T, Steinheim, Germany) for 1 h at 37 °C, with shaking. We removed the excess conjugate by washing it three times with TBS-T. Immune complexes were stained with NBT (nitrotetrazol) and BCIP (5-Bromo-4-chloro-3-indolyl-phosphate) substrates (Merk, B5655, Darmstadt, Germany) for alkaline phosphatase, diluted in TBS containing Mg^2+^ ions, pH 9.5. After 15–20 s, we stopped the reaction by washing with water.

### 2.6. Isolation of Phage Protein via Preparative Electrophoresis in Polyacrylamide Gel

We separated the proteins with a Prep Cell kit (Model 491 from Bio-Rad #1702926, Bio-Rad in Europe, Hercules, CA, USA). The gradient separation gel contained 5, 8, 10, and 12.5% (*w/v*) layers. The sample with whole bacteriophage particles was sonicated for 10 min in the ultrasonic disintegrator, MSE apparatus. Then, we added one volume of denaturing buffer (concentrated 2 times) and heated the proteins for 10 min at 100 °C. Proteins were separated in a discontinuous system, using a Tris/Gly electrode buffer, pH 8.3 (Thermos Scientific, 15413679, Waltham, MA, USA). The current voltage was 260 V, and fractions (1.5 mL) were eluted into the water at a rate of 1 mL in 1.5 min. The electrophoresis took 30 h. The collected proteins were used for immobilization to Sepharose-4B for affinity chromatography.

### 2.7. Preparation of Antibodies by Affinity Chromatography

Human IgG antibodies were isolated from serum, using affinity chromatography to separate phage proteins. We prepared the chromatographic column by immobilization of bacteriophage proteins to Sepharose-4B. To the 10 mL of the Sepharose-4B gel (Pharmacia, 17–0756–01), 4 M NaOH (POCH, 1310-73-2, Polish Chemical Reagents, Gliwice, Poland) was added to obtain a pH of 11. Then, we added 1 g of CNBr (Cyanogen bromide) (Sigma-Aldrich, 820194, Steinheim, Germany) and kept the pH constant in the 10–11 range. The gel was washed with cold water on a glass filter and then with 0.1 M carbonate buffer pH 8.6 (IITD, PAS), and incubated with 30 mg of phage proteins (0.1 M carbonate buffer pH 8.6), gently mixed for 3 h. After that, two volumes of ethanolamine (pH 8.0) were added and incubated for 3–5 h, with constant stirring at room temperature, 4 °C overnight. We washed the gel with 2 M K_2_HPO_4_ (IITD, PAS), then with water and Phosphate Buffered Saline (PBS).

Immunoglobulins were prepared from human serum, using 40% (*w/v*) ammonium sulfate (Sigma-Aldrich, A4418, Steinheim, Germany) precipitation [14]. We added a saturated solution of ammonium sulfate (1 g/mL) to the serum in the ratio of 2:1 (*v/v*), respectively. After incubation at 4 °C for 1 h with occasional mixing, the immunoglobulins were centrifuged at 1000× *g* at 4 °C for 15 min. After that, the pellet was dissolved and dialyzed against PBS overnight. The immunoglobulins were concentrated on Amicon concentrators, using a 100 kDa membrane.

Affinity chromatography was performed slowly (1 drop/15 min), and we washed out the unbound fraction with PBS. Antibodies were eluted with 1 M NaCl in PBS, then with 3 M KSCN (Pol-Aura, 117455008#25KG, Olsztyn, Poland) (Potassium thiocyanate) in PBS, and finally were dialyzed to the PBS, at 4 °C, overnight. In the next step, we concentrated the antibodies by Amicon ultrafiltration and stored them at −20 °C in 50% glycerol.

### 2.8. ELISA

We used Costar 3596 plates coated with bacteriophage proteins of 1 µg/well. The plates were incubated at 4 °C overnight and blocked with 1% BSA solution in TBS-T buffer, pH 7.5. We performed the interaction with human sera (diluted 250 times) or human anti-phage antibodies (1 μg) for 2.5 h at room temperature. Then, the plate was washed with TBS-T, and the reaction with the anti-human IgG conjugated with alkaline phosphatase was performed. We measured the absorption at λ = 405 nm on a BIOTEK Power Wale XS Microplate Spectrophotometer once the colorimetric reaction was developed. Each sample was triplicate, and the average value was calculated. We analyzed the values statistically, using a one-way analysis of variance (ANOVA) or the Shapiro–Wilks test, with the GraphPad Prism software package (www.graphpad.com, accessed on 1 July 2014).

## 3. Results

We purified phages I11mO19, T4, ΦK1E, and F8, using methods described by Lipiński et al. [11]. The titer was 2.8 × 10^13^, 9.3 × 10^13^, 4.0 × 10^12^, and 3.0 × 10^12^ PFU for I11mO19, T4, ΦK1E, and F8 phage, respectively. All samples were free from bacterial lipopolysaccharide (LPS). The level of endotoxin was <50 EU/mL for all phage samples. The protein patterns of all studied phages differed from those of *E. coli* hosts. However, some bands detected in both the phage and hosts’ samples shared the same molecular weight (Appendix A).

In the beginning, we analyzed the human sera of the donors untreated with the phages to assess whether there was any immunoreactivity to phage I11mO19. Therefore, we performed a WB analysis. Once we noticed the reactivity for some individuals’ sera, we used the reactive sera in ELISA to evaluate the level of reactivity. In all immunoenzymatic tests, we tested 150 human sera. However, in the WB, we used 46 of them. Thirty-eight sera were immunoreactive to I11mO19 phage proteins. The reactivity varied, and the differences—proteins and intensity of interactions—were observed. The representative results of phage proteins’ immunoreactivity are shown in Figure 1.

The test showed the diversity of human sera in terms of reactivity with the 11mO19 phage proteins. Some of the analyzed sera gave a positive result with almost the entire protein panel (>10 proteins) of the I11mO19 bacteriophage (sample A serum in Figure 1). Some sera were negative with the phage proteins (serum E), while some sera reacted only with a few proteins (<10).

To determine the amount of anti-phage antibodies, we used a quantitative ELISA. However, we first had to isolate specific anti-I11mO19 antibodies from human sera determined as positive. Then, we used these antibodies to create a standard curve for anti-I11mO19 antibody determination in analyzed sera. The phage-specific antibodies were isolated from human sera, using affinity chromatography. We separated phage proteins in preparative SDS-PAGE, followed by immobilization to the Sepharose-4B beads, with a yield of up to 94%. To precipitate immunoglobulins, we used pooled serum derived from 20 blood donors, and then we subjected the IgG fraction to affinity chromatography. Ten percent of the precipitated IgG antibodies bound to the phage proteins. We eluted and neutralized the antibodies, after which we evaluated immunoreactivity evaluation in WB (Figure 2).

These purified antibodies showed similar reactivity to I11mO19 phage proteins as human sera, meaning the antibodies were still active even though harsh elution agents were used (KSCN). We analyzed these anti-I11mO19 antibodies in WB with all purified phages (Figure 3).

We used the isolated anti-I11mO19 antibodies to assess cross-reactivity to T4, ΦK1E, and F8 bacteriophages. We noticed a cross-reactivity of the isolated anti-I11mO19 antibodies with T4 phage proteins and a weak interaction with ΦK1E phage (Figure 3A and Appendix A). We also analyzed the immunoreactivity of isolated anti-I11mO19 IgGs to *Pseudomonas* phage F8 proteins (Figure 3B), and we did not observe any positive reaction in WB. Anti-I11mO19 antibodies seemed to be specific only to coli phage proteins, not *Pseudomonas* phage F8.

The next task concerned elaborating the ELISA parameters to assess the level of the anti-phage antibodies in the individual sera samples. For this purpose, we first collected 24 sera determined as negative in WB, and we used the anti-I11mO19 purified antibodies as a reference. These non-reactive sera allowed us to draw an arithmetic mean and standard deviation (SD) for the negative sera and allowed us to estimate the limit of reactivity for human serum under the conditions of the ELISA (Appendix A).

We performed the same analysis for the reference anti-I11mO19 antibodies and presented the reactivity profiles in Figure 4. We used these calibration curves as a criterion for classifying the study of human sera as an immunoreactive or non-immunoreactive one. Sera were diluted four-fold and considered negative when the absorbance value at λ = 405 nm (A405) was ≤0.25 at the test conditions. Then, we examined immunoreactive sera to determine the level of anti-I11mO19 antibodies.

Most of the reactive sera’ curves were parallel (in the range of dilutions ¼ to ^1/^_32_) to the reference line, allowing us to use the standard curve to determine the level of serum antibodies. We tested one hundred twenty human sera in ELISA, and 27% were negative. To determine the amount of anti-I11mO19 antibodies, we calculated A405 at the test conditions for the reaction of phage proteins, with serum diluted eight times. We calculated the level of immunoglobulins in a serum sample based on the A405 values, taken from the standard curve. For a dilution of ^1^/_8_ of “standard”, a value A405 was 0.315, corresponding to 16 µg/mL protein in the diluted sample (128 µg/mL in undiluted sera). Among the presented 88 (N = 88) immunoreactive sera, 22 (25%) contained an antibody level between 100 and 200 µg/mL, and the antibody level in the case of 61 (70%) sera was over 200 µg/mL, and 4 (5%) sera contained above 400 µg/mL of anti-I11mO19 antibodies.

We also carried out an ELISA experiment to confirm the difference in reactivity of individual human sera with the phage proteins (Figure 5).

The results of the ELISA test coincided with the immunoblotting results for those sera analyzed in both tests. Most of the sera gave a positive reaction in the ELISA test. However, among 88 tested sera, five gave a result much higher than the other sera (Figure 5). λ = A405 nm for these samples was higher than 0.4.

## 4. Discussion

In this manuscript, we present the results concerning our study on the presence of anti-phage antibodies in human sera naturally created during a lifetime. This issue is essential regarding the potential neutralization effect of the antibodies on phage and the unfavorable impact on phage therapy [15,16]. In this manuscript, the naturally occurring antibodies state was that of antibodies generated during the lifetime of individuals without purpose-targeted immunization. These antibodies were mainly IgG and are thought to be produced by “conventional” (CD5-) B cells [17]. However, the investigation of this activity relates to many methodological problems. First, to gain reliable results, we prepared bacteriophages free of bacterial contamination to exclude false-positive results. For example, the healthy human serum contains antibodies against outer membrane proteins of the Enterobacteriaceae family, mainly to protein OMP [18]. The preparation of pure bacteriophages is complex, not only because they are in heterogeneous mixtures of molecules but also often due to their irreversible interactions with bacterial receptors. Thus, our first goal was to obtain phage preparations of immunological purity. Standard methods for bacteriophage purification involved the following stages: ammonium sulfate or polyethylene glycol precipitation [15,19]; ultracentrifugation in saccharose or cesium chloride gradients [16]; and chromatographic techniques [19,20].

Another problem was the analytical methodology for phage purity assessment. Low amounts of anti-phage antibodies in serum cause problems when a routine test is used for their determination. Therefore, this work attempted to solve these problems in order to prove unequivocally the presence of naturally occurring anti-phage antibodies and to develop an assay for anti-phage antibodies’ detection. There are only a few reports on this topic, all suffering from the above-mentioned methodological concerns [6,21,22]. In this work, we wanted to show whether assessing the level of phage-specific antibodies naturally occurring in human blood is possible. We chose I11mO19 bacteriophage due to the high incidence of its host in the Polish population in the recent decade. We also intended to assess the cross-reactivity anti-phage antibodies to *E. coli* phages T4 and ΦK1E. In principle, we have proved the earlier data from our team indicating that, indeed, phage-specific antibodies can occur in the sera of healthy individuals. Immunologically pure phage particles devoid of host structures allowed revealing the existence of natural anti-phage antibodies in their diverse quantities unequivocally.

Phage particles were purified using DOC and EDTA reagents to prevent the association with bacterial cell fragments of newly released phages. The reagents were added shortly before the completion of the lysis. These compounds caused a 60% and 54% increase in phage titers. DOC and EDTA are potent in destroying aggregated fragments of bacterial cells and preventing them from binding to the phage particles [5]. The developed method of phage purification presented in this article was relatively short and efficient. The purity of phage preparations was analyzed by determination of endotoxin, using the LAL assay, and its amounts were <50 EU/mL (the endotoxin content of distilled water was 20 EU/mL [22]. Purity was also assessed by SDS-PAGE compared to bacterial proteins and revealed that major bacterial proteins were not in the phage samples.

We assessed the immunoreactivity of I11mO19 phage proteins and showed differences in the reactivity patterns among analyzed human sera. (This analysis followed the assessment of differences in reactivity between host proteins and the phage particles with human serum—Appendix A) We noticed the cross-reactivity of anti-I11mO19 antibodies with other *E. coli* phages. However, the antibodies did not react with *Pseudomonas* phage proteins. Antibodies’ cross-reactivity depended on the antigen’s similarity, so the reactive coliphage proteins may share the same epitopes. The diversity among E coli infecting phages is known to be highly widespread. Comparisons of the T4-type genomes have shown that these phages share a typical ancestral genome but have been modified in numerous ways during their evolution. Generally, the phages that infect closely related host bacterial species are phylogenetically closer to each other than those that infect distantly related hosts. Therefore, we can expect the homology between coliphage proteins to exist [23]. In the studies, the cross-reactivity of antibodies anti-coliphage occurs mainly against major capsid phage antigens. For example, the major capsid protein of the T4 phage (gp23) is identical in over 60% with significant capsid proteins of about 125 other coliphages (based on BLAST analysis). The receptor for the T4 phage involves wild-type LPS. Therefore, this phage may recognize a broader spectrum of bacteria [7]. Phage ΦK1E is specific for *E. coli* K1, a human pathogen, with capsular colominic acid as a receptor, the structure is also present in human polysialoglyco proteins [24,25,26]. This phage may interact with human tissues expressing colominic acid on the cell surface. The presence of the host bacteria in the human organism indirectly indicates the presence of a phage that infects the bacteria strain.

First, the initial assumption was to assess the presence of specific antibodies to phage proteins in human sera of non-immunized with phage, healthy individuals. We performed a WB analysis, based on which, we could determine the presence of immunoreactivity (it was not a quantitative analysis). Based on the WB assay, we grouped sera as reactive and non-reactive. Since we did not have any commercial antibodies that we could use as a standard, our strategy was, at first, to assess the immunoreactivity of each serum to phage proteins, then isolate the antibodies and use them as a standard. Once we saw diversity in reactivity, we decided to group the sera as reactive and non-reactive. The most positive sera were pooled and used in affinity chromatography to isolate phage-specific antibodies. We used these purified antibodies as a standard in ELISA.

We developed the ELISA to determine the level of anti-phage antibodies. Selected I11mO19 phage is rare; therefore, we expected distinct individual differences in antibody levels, which could facilitate the interpretation of results. The obtained results fit the immunoblotting results for not-so-high sera (from ¼ to ^1/^_32_). The non-reactive immunoblotting sera allowed the creation of an accurate profile of the negative serum in ELISA, a reference profile to that positive one, drawn for purified antibodies. It appeared that profiles for the immunoreactive sera were similar to the reference antibodies’ profile, allowing us to determine actual amounts of anti-phage antibodies in sera. On average, the concentrations of anti-I11mO19 antibodies were 197 µg/mL. This IgG antibody amount included all pools of immunoglobulins against all I11mO19 phage proteins, and, additionally, against several epitopes of the antigens. When the level of IgG immunoglobulins in the healthy human organism reaches 1.5 mg/mL [27], the determined quantity of the anti-I11mO19 phage is 13% of the total IgG. Considering the cross-reactivity, we assume that not only the anti-I11mO19 antibodies have been recognized (in the ELISA conditions), but also other anti-coliphages, or even the antibodies against the phages of Enterobacteriaceae. We should take into account the cross-reactivity of anti-phage antibodies with bacterial epitopes. As we can assume, when administered as any other antigen, phages can trigger the immune system for specific antibody production, as already determined [28,29,30]. However, the study for anti-phage antibodies in non-treated phages individuals has yet to be studied, and the first try has been presented in this paper.

## 5. Conclusions

We have confirmed that healthy individuals produce anti-coliphage antibodies, and ELISA can estimate the quantity using the isolated anti-phage IgG antibodies as a standard. In this work, we assessed the amount of anti-I11mO19 phage antibodies. Although they are also specific to other *E. coli* phages, they did not react with *Pseudomonas* bacteriophage proteins. This work is novel because it proposes a new and valuable ability to develop a helpful assay for specific anti-phage antibody determination. This assay is a good starting point for developing an individual and more thorough test for determining anti-phage antibodies in human sera against bacteriophages. This could contribute to further studies on their significance in human health and disease and their importance in the outcomes of phage therapy.

## Figures and Tables

**Figure 1 antibiotics-12-00586-f001:**
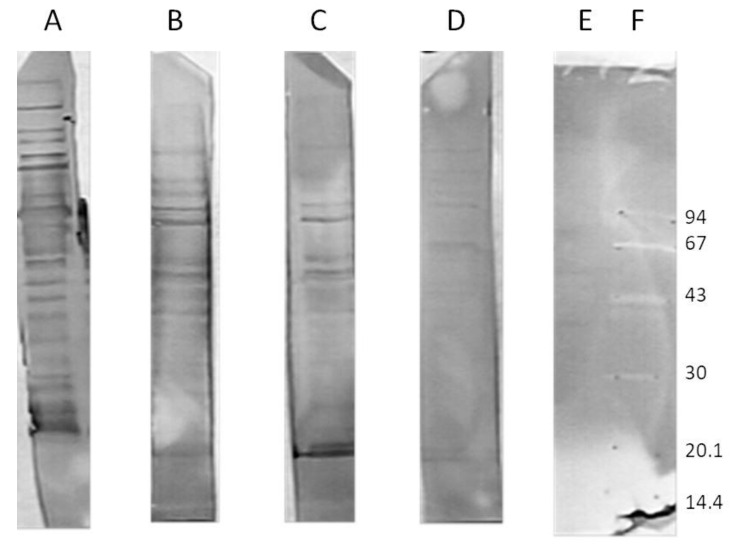
Immunoreactivity of different human sera to I11mO19 phage proteins in immunoblotting analysis (**A**–**E**), (**F**) molecular weight standard (kDa). Sera were diluted 250 times. Anti-human IgG antibodies labeled with alkaline phosphatase were used in this test. The secondary antibodies were diluted 10,000 times.

**Figure 2 antibiotics-12-00586-f002:**
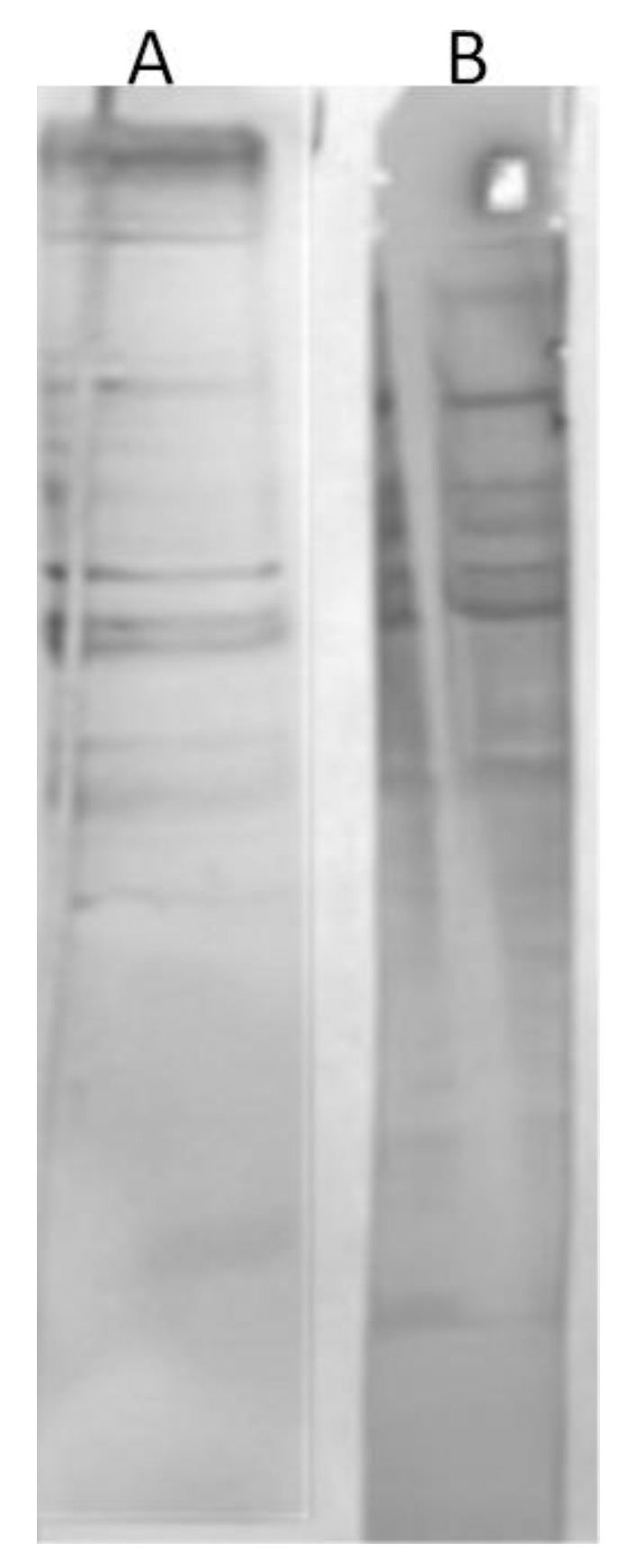
Representation of the comparative analysis of immunoreactivity between anti-I11mO19 antibodies obtained in affinity chromatography (**A**) versus a positive human serum (**B**). In the WB test, 30 µg of isolated antibodies, 250 times diluted serum, and goat anti-human antibodies IgG labeled, with alkaline phosphatase, diluted to 1/10,000, were used.

**Figure 3 antibiotics-12-00586-f003:**
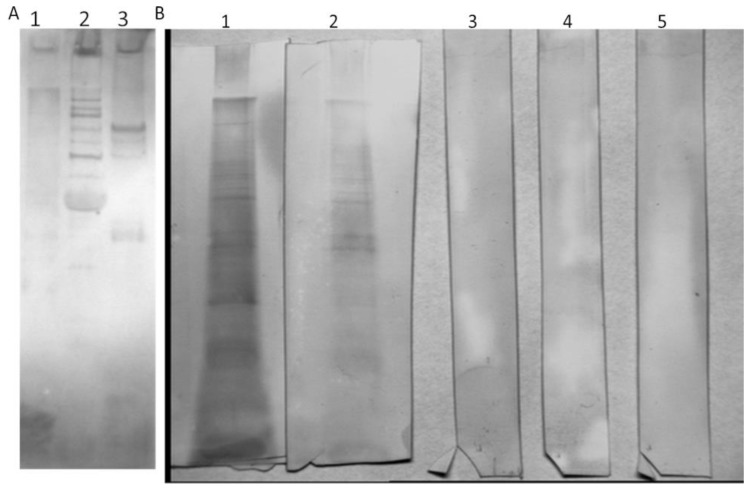
Immunoreactivity analysis of isolated anti-I11mO19 antibodies phage proteins and (**A**) lane 1 ΦK1E, lane 2 T4, lane 3 I11mO19 phage proteins. The anti-I11mO19 antibodies were not reactive to F8 phage proteins (**B**) lane 5. Neither antibodies specific to the T4 phage nor specific to ΦK1E interacted with the F8 phage (**B**) lane 3 and 4, respectively. In contrast, human sera and isolated anti-F8 antibodies reacted with the F8-phage proteins (**B**) lanes 1 and 2, respectively. In the WB test, 30 µg of isolated antibodies and secondary anti-human IgG antibodies, labeled with alkaline phosphatase, were used. The secondary antibodies were diluted 10,000 times.

**Figure 4 antibiotics-12-00586-f004:**
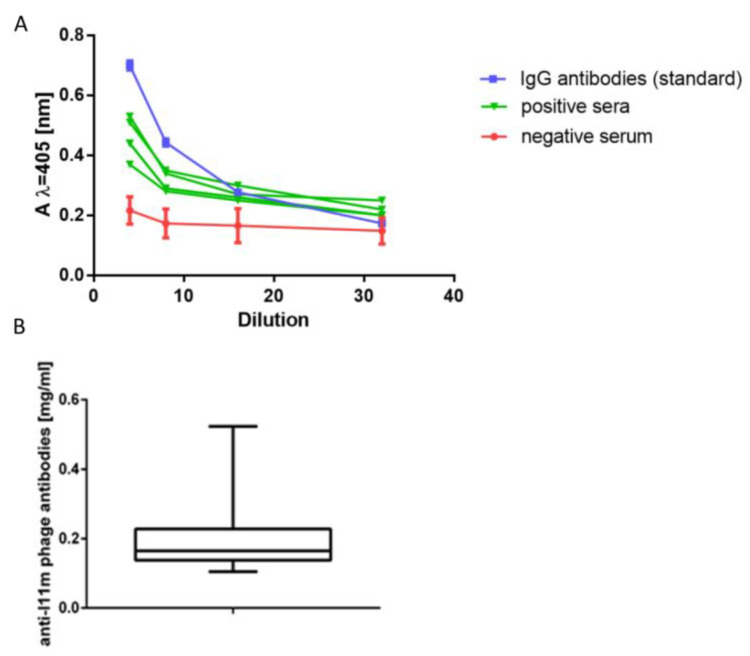
(**A**) Profiles of reactivity in ELISA for the negative (red) and the immunoreactive sera (green) with proteins of I11mO19 phage, compared to the reference profile (blue) of the affinity-purified 11mO19 antibodies. (**B**) Quantitative representation of anti-phage antibodies in the human population. The mean value was 0.197 mg/mL (N = 88, SD = 0.09134, *p* < 0.0001, Shapiro–Wilks). The profile for negative serum presents the mean value of 24 non-reactive sera tested in the WB.

**Figure 5 antibiotics-12-00586-f005:**
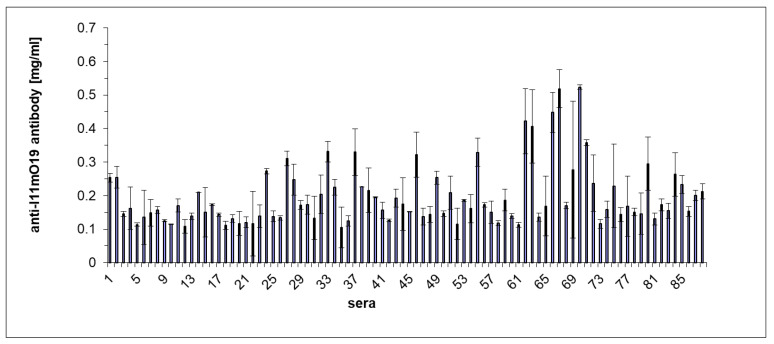
ELISA showing immunoreactivity of human sera with the proteins of the bacteriophage I11mO19. Sera were diluted 250 times, and the plate was coated with phage proteins at 1 µg/well. The alkaline phosphatase labeled anti-human IgG conjugate was diluted 10,000 times.

## Data Availability

Not applicable.

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
