# Peer review of "Detection and Level Evaluation of Antibodies Specific to Environmental Bacteriophage I11mO19 and Related Coliphages in Non-Immunized Human Sera"

_antibiotics, 2023, doi:10.3390/antibiotics12030586_

Round 1
Reviewer 1 Report
General comment
This is a fine piece of work that deserves to be published by this journal after the below minor issues have rectified.
Minor comment
1. Introduction
L59 & L63: Italicize this word “E. coli”.
L64: Italicize these word “Pseudomonas”.
2. Materials and Methods
2.2 Bacteriophages propagation and purification
L79: State the source of this media; giving the company name, its location (city & country) or provide the product's catalogue number.
L80: Is this 109 CFU/mL or 10^9 CFU/mL? I doubt whether 109 CFU of bacteria in 1 mL will give an O.D of 1.00!
L81: Provide more details regarding the source of this centrifuge giving the company name, its location (city & country) or provide the product's catalogue number.
L81: Provide more details regarding the source of this sterile tubes giving the company name, its location (city & country) or provide the product's catalogue number.
L83-84: State the source of this media; giving the company name, its location (city & country) or provide the product's catalogue number.
L85: State the source of this media; giving the company name, its location (city & country) or provide the product's catalogue number.
L89: State the source of this media; giving the company name, its location (city & country) or provide the product's catalogue number.
L95-96: State the source of this device; giving the company name, its location (city & country) or provide the product's catalogue number.
L100: Clearly state the manufacturer of this enzyme; giving the company name, its location (city & country) or provide the product's catalogue number.
L102: State the source of this product; giving the company name, its location (city & country) or provide the product's catalogue number.
2.3 Determination of the LPS Content
L115: Italicize this word “E.coli”.
L115-116: State the source of this endotoxin; giving the company name, its location (city & country) or provide the product's catalogue number.
2.5 Immunoblotting
L130: State the source of this product; giving the company name, its location (city & country) or provide the product's catalogue number.
L132: State the source of this product; giving the company name, its location (city & country) or provide the product's catalogue number.
L141: State the source of this product (NBT); giving the company name, its location (city & country) or provide the product's catalogue number.
L141: State the source of this product (BCIP); giving the company name, its location (city & country) or provide the product's catalogue number.
2.4 Isolation of phage protein via preparative electrophoresis in polyacrylamide gel
L147: State the source of this equipment; giving the company name, its location (city & country). Alternative provide the product's catalogue number.
L152: State the source of this product; giving the company name, its location (city & country) or provide the product's catalogue number.
2.5 Preparation of antibodies by affinity chromatography
L158-159: State the source of this product; giving the company name, its location (city & country) or provide the product's catalogue number.
L162: State the source of this product; giving the company name, its location (city & country) or provide the product's catalogue number.
L174: State the source of this product; giving the company name, its location (city & country) or provide the product's catalogue number.
2.6 ELISA
L178: Find a clear way of explaining how you (the authors) arrived to this number (10^7) of phage particles. Did you intend to imply 10^7 pfu/mL?
3. Results
L255, L257: Italicize this word “Pseudomonas”.
L305: what could be the reason for the high positive reactions observed with the five sera in Figure 5?
4. Discussion
L336: grammatical error. "normindividuals" instead of "normal individuals".
Author Response
Dear Reviewer, We are very grateful for your comments. We added all the missing data you suggested.
General comment
This is a fine piece of work that deserves to be published by this journal after the below minor issues have rectified.
Minor comment
- Introduction
L59 & L63: Italicize this word “E. coli”.
It has been corrected
L64: Italicize these word “Pseudomonas”.
It has been corrected
- Materials and Methods
2.2 Bacteriophages propagation and purification
L79: State the source of this media; giving the company name, its location (city & country) or provide the product's catalogue number.
It has been added
L80: Is this 109 CFU/mL or 10^9 CFU/mL? I doubt whether 109 CFU of bacteria in 1 mL will give an O.D of 1.00!
It has been corrected
L81: Provide more details regarding the source of this centrifuge giving the company name, its location (city & country) or provide the product's catalogue number.
It has been added
L83-84: State the source of this media; giving the company name, its location (city & country) or provide the product's catalogue number.
It has been added
L85: State the source of this media; giving the company name, its location (city & country) or provide the product's catalogue number.
It has been added
L89: State the source of this media; giving the company name, its location (city & country) or provide the product's catalogue number.
It has been added
L95-96: State the source of this device; giving the company name, its location (city & country) or provide the product's catalogue number.
It has been added
L100: Clearly state the manufacturer of this enzyme; giving the company name, its location (city & country) or provide the product's catalogue number.
It has been added
102: State the source of this product; giving the company name, its location (city & country) or provide the product's catalogue number.
It has been added
2.3 Determination of the LPS Content
L115: Italicize this word “E.coli”.
It has been corrected
L115-116: State the source of this endotoxin; giving the company name, its location (city & country) or provide the product's catalogue number.
The endotoxin was included in the kit from Lonza. The cat. Number was added.
2.5 Immunoblotting
L130: State the source of this product; giving the company name, its location (city & country) or provide the product's catalogue number.
It has been added
L132: State the source of this product; giving the company name, its location (city & country) or provide the product's catalogue number.
It has been added
L141: State the source of this product (NBT); giving the company name, its location (city & country) or provide the product's catalogue number.
It has been added
L141: State the source of this product (BCIP); giving the company name, its location (city & country) or provide the product's catalogue number.
NBT/BCIP were combined together. The cat. number was added
2.4 Isolation of phage protein via preparative electrophoresis in polyacrylamide gel
L147: State the source of this equipment; giving the company name, its location (city & country). Alternative provide the product's catalogue number.
It has been added
L152: State the source of this product; giving the company name, its location (city & country) or provide the product's catalogue number.
It has been added
2.5 Preparation of antibodies by affinity chromatography
L158-159: State the source of this product; giving the company name, its location (city & country) or provide the product's catalogue number.
It has been added
L162: State the source of this product; giving the company name, its location (city & country) or provide the product's catalogue number.
It has been added
L174: State the source of this product; giving the company name, its location (city & country) or provide the product's catalogue number.
It has been added
2.6 ELISA
L178: Find a clear way of explaining how you (the authors) arrived to this number (10^7) of phage particles. Did you intend to imply 10^7 pfu/mL?
There was a mistake; the plates were coated with 1 µg/well phage proteins. It has been corrected.
- Results
L255, L257: Italicize this word “Pseudomonas”.
It has been corrected
L305: what could be the reason for the high positive reactions observed with the five sera in Figure 5?
In our opinion. These five individuals had higher levels of anti-coliphage antibodies. Since there was a probability they had contact with the phages, which were considered population phages
- 4. Discussion
L336: grammatical error. "normindividuals" instead of "normal individuals".
It has been corrected
Reviewer 2 Report
This is a simple paper that looks at sera from a number of human samples and asks whether antibodies to specific phage (or more accurately phage proteins) exist in those sera. These are somewhat random samples from individuals not known to be exposed to the phages being used.
Fundamentally the study is ok although not very significant, however I think a few controls are missing. As one might expect, a signficant sample of the sera do show reactivity against the test phage and a signifiant number of samples do not.
I have two primary concerns with the work. First, I have not seen appropriate controls to show me the proteins being detected are in fact phage proteins and not host proteins. If their phage has been CsCl gradient purified or some other "gold standard" purification then I would be ok, but I do not see evidence their phage samples do not have host proteins (I can accept they do not have host peptidoglycan).
Secondly, in figures 1-3 they show multiple strips from western blots. Are these all from the same original gel and membrane or are they from different blots. Hopefully the former since it isn't really appropriate to compare different blot results unless you have an identical positive control lane on all membranes.
I would have liked to see a lane with only a bacterial extract as a control as well for comparison.
In the elisa (Fig 5) was there any control wells?
The most obvious and perhaps relevant experiment was not done by the authors. As they state the relevance of antibodies in human serum is only if they inactivate the phage and thereby render phage therapy approaches problematic, as they state in lines 310-311. Given that they have phage and sera, why did they not test whether any of the human sera neutralized the phage?
Line 356 what does normindividuals mean? or is this an error.
There are some issues with the legends in the Supplementary text . Legend to Fig 1B is confusing and there do not seem to be any descriptions of the lanes in Fig 2 at all.
In summary the manuscript needs a few more controls to be acceptable (or better explanations as to why they are not necessary) but the major issue is that there is nothing significantly new in the manuscript. By their own words (line 379) many studies have already shown human sera have antibodies against phage. I don't see anything especially new here nor do I see much relevance unless they can also demonstrate these sera are neutralizing. Otherwise the finds are immaterial.
Author Response
Dear Reviewer, We are very grateful for your comments. Please find our response to criticisms and comments below.
Rev. 2.
This is a simple paper that looks at sera from a number of human samples and asks whether antibodies to specific phage (or more accurately phage proteins) exist in those sera. These are somewhat random samples from individuals not known to be exposed to the phages being used.
The study aimed to check whether a normal, non-immunized with phage, healthy person has anti-phage antibodies. The question is fundamental regarding potential contact with phages, the possibility of having a natural anti-phage response, how this response is cross-reactive with groups of phages, the relationship to antibacterial immunity, and the practical goal regarding phage therapy.
We have chosen a phage specific to the clinical bacterial strain, the most popular in the population. Still, at the same time, two other coliphages were analyzed, well-described reference entities that could be considered controls.
To prevent even trace contaminants, we have purified them in 3 separate new pellicon devices. Additional control was Pseudomonas phage, which has led us to an exciting conclusion on some similarities between coliphages, utterly different from Pseudomonas. We have encountered severe difficulty with low amounts of such antibodies in human serum. The solution to this problem appeared in the screening experiment with Western blotting on many serum samples, which allowed us to visualize the reactivity with three coli phage proteins. Then it was possible to elaborate the ELISA assay using highly reactive and non-reactive sera treated as positive and negative samples, respectively. With the development of this assay, further work is possible. The procedure elaborated allowed us to monitor the purity of phage particles devoid of bacterial substances with several methods to consider phage preparations as immunologically pure. Purity was the primary condition in these experiments, controlled using several tests. The results presented in this manuscript will be significant in the future concerning microbiome studies, mechanisms of infectious diseases, phage therapy, immunology of infectious diseases, and other fields.
Fundamentally the study is ok although not very significant, however I think a few controls are missing. As one might expect, a signficant sample of the sera do show reactivity against the test phage and a signifiant number of samples do not.
I have two primary concerns with the work. First, I have not seen appropriate controls to show me the proteins being detected are in fact phage proteins and not host proteins. If their phage has been CsCl gradient purified or some other "gold standard" purification then I would be ok, but I do not see evidence their phage samples do not have host proteins (I can accept they do not have host peptidoglycan).
In the presented study, we used a continuous ultrafiltration method for bacteriophage particle purification. All phages described in this paper use polysaccharides as receptors to reach the bacterial cell. Therefore, in this case, the purity determinant was the level of endotoxin (lipopolysaccharide) in purified phage samples. We monitored the purification process with the LAL test until endotoxin was removed from phage particles in the presence of detergent, which is described in the text of the manuscript.
The purification methodology assumed that the presence of DOC and EDTA reagents prevents the association with bacterial cell fragments and destroys aggregates of bacterial debris, preventing binding to them the phage particles (This is also included in the Discussion section) . We assumed that all bacterial contamination, including proteins, was removed during continuous ultrafiltration, where we used a 1000K kDa cut-off membrane. Theoretically, only phage particles should be on hold by this membrane type. However, there was a minimal risk that the host's proteins were smuggled with the phage particle. To compare the panel of proteins of phages and their hosts, we conducted a comparative analysis of the SDS-PAGE, and the details are presented in the supplementary material in Figure S1 S2, Table S1 S2
Secondly, in figures 1-3 they show multiple strips from western blots. Are these all from the same original gel and membrane or are they from different blots. Hopefully the former since it isn't really appropriate to compare different blot results unless you have an identical positive control lane on all membranes.
We have done several experiments, and these presented are representative ones. However, the circumstances of the experiment were not standard. We performed immunoblotting of phage proteins with different human sera as follows: after phage proteins transferred on the membrane, we cut it, forming strips for incubation with varying sera of humans into various troughs. The experiment's conditions were the same for all blots, so the strips come from the same experiment In Figure 1, the analysis concerns band visibility (Y/N analysis) and noticing the potential differences between the protein's reactivity. We analyzed all sera under the same conditions, around 20 sera in the same experiment (due to physical feasibility limitations, we were able to explore approximately 20 sera in one experiment). There was none of the appropriate positive control available. However, we included the comparative analysis of immunoreactivity between phage I11mO19 and the host's proteins (together with LPS) using human serum (Supplementary material Figure S5). In Figures 2 and 3, we performed immunoblotting using isolated antibodies, so they can be assumed to be a positive control
I would have liked to see a lane with only a bacterial extract as a control as well for comparison.
We added the comparative analysis of immunoreactivity between phage I11mO19 and the host's proteins (together with LPS) using human serum (Supplementary material Figure S5).
In the elisa (Fig 5) was there any control wells?
The ELISA assay has been performed with all standard controls, namely for antigen-coated cross-reactivity, reagents, and a secondary antibody controls
The most obvious and perhaps relevant experiment was not done by the authors. As they state the relevance of antibodies in human serum is only if they inactivate the phage and thereby render phage therapy approaches problematic, as they state in lines 310-311. Given that they have phage and sera, why did they not test whether any of the human sera neutralized the phage?
This intriguing question is fundamental and arises from our studies. The primary goal was reached with our research; we have found the existing natural anti-phage antibodies in human serum are at various levels. This is the first document in this issue, as far as we know. The biological role of these antibodies is the topic of further investigation, and the presence of possible immune complexes is a real challenge.
Line 356 what does normindividuals mean? or is this an error.
It has been corrected
There are some issues with the legends in the Supplementary text . Legend to Fig 1B is confusing and there do not seem to be any descriptions of the lanes in Fig 2 at all.
It has been corrected
In summary the manuscript needs a few more controls to be acceptable (or better explanations as to why they are not necessary) but the major issue is that there is nothing significantly new in the manuscript. By their own words (line 379) many studies have already shown human sera have antibodies against phage. I don't see anything especially new here nor do I see much relevance unless they can also demonstrate these sera are neutralizing. Otherwise the finds are immaterial.
The manuscript combines results in a shorter, comprehensive way of a more significant sequence of experiments, with reliable controls, for example, purification with salt or saccharose gradients, glycol precipitation purification, and finally, pellicon purification which gave the most effective, less time-consuming, simplified and cost-effective procedure, patented by our team (ref. 9, US Patent 2011/0008873 A1). The cited publications only tentatively described the presence of anti-phage antibodies detected with whole phage particles, but these experiments were necessary to be proved. What is distinguished in this presented study is that the Immunologically pure phage particles devoid of host structures allowed revealing the existence of natural anti-phage antibodies in their diverse quantities unequivocally. Moreover, the study for anti-phage antibodies in non-treated phages individuals has yet to be studied, and the first try is presented in this paper. We have improved the manuscript accordingly in the new version.
Reviewer 3 Report
This is a nice paper demonstrating the method to detect anti-phage antibody in human sera which is an important research field to facilitate the use of phage therapy. There are a few things that required attention as stated below:
Page 5 and 6, Figure 1 and 3: The clarity can be improved. Legend description can be improved to make it easier to follow.
Page 5, lines 198 – 201: The statements are confusing as it started from different reactivity to I11mO19 then cross-reactivity to other phages but the following statements move back to the difference in reactivity to only I11mO19.
Page 5, lines 224 – 225: The authors mentioned that they isolated specific anti- I11mO19 antibodies; however, it is not clear to me which 20 human sera were used and why they were chosen. I think this is an important factor as the interpretation on the downstream data depends on it.
It would be great to include the information on the different coliphages to show how closely related they are and perhaps this can lead to the identification of determinants causing the cross-reactivity.
Minor nomenclature error (E. coli is not in italic format)
Author Response
Dear Reviewer, We are very grateful for your comments. Please find our response to criticisms and comments below.
Rev. 3
This is a nice paper demonstrating the method to detect anti-phage antibody in human sera which is an important research field to facilitate the use of phage therapy. There are a few things that required attention as stated below:
Page 5 and 6, Figure 1 and 3: The clarity can be improved. Legend description can be improved to make it easier to follow.
It has been corrected
Page 5, lines 198 – 201: The statements are confusing as it started from different reactivity to I11mO19 then cross-reactivity to other phages but the following statements move back to the difference in reactivity to only I11mO19.
It has been corrected
Page 5, lines 224 – 225: The authors mentioned that they isolated specific anti- I11mO19 antibodies; however, it is not clear to me which 20 human sera were used and why they were chosen. I think this is an important factor as the interpretation on the downstream data depends on it.
First, the initial assumption was to assess the presence of specific antibodies to phage proteins in human sera of non-immunized with phage, healthy individuals. We performed a WB analysis based on which we could determine the presence of immunoreactivity (it was not a quantitative analysis). Based on the WB assay, we grouped sera as reactive and non-reactive. Since we didn’t have any commercial antibodies that we could use as a standard, our strategy was, at first, to assess the immunoreactivity of each serum to phage proteins, then isolate the antibodies and use them as a standard. Once we saw diversity in reactivity, we decided to group the sera as reactive and non-reactive once. The most positive sera were pooled and used in affinity chromatography to isolate phage-specific antibodies. We added the above annotation to the Discussion section.
It would be great to include the information on the different coliphages to show how closely related they are and perhaps this can lead to the identification of determinants causing the cross-reactivity.
The research covered bacteriophages: T4, ФK1, and I11m, whose hosts are strains of Escherichia coli bacteria, differing in surface receptors. The investigation began with the T4 phage, which infects E. coli B cells, and the receptor is a rough-type lipopolysaccharide on the surface of the bacterial cell. The T4 phage is well known and widely described in the literature, a model phage for laboratory work, which was to facilitate the development of methods for the propagation and purification of both this and other phage particles, also crucial for humans. Another phage model selected for the study is the ФK1 phage infecting E. coli K1 strains, and the receptor is the capsular polysaccharide, polysialic acid called colominic acid. The Escherichia coli K1 bacterium is pathogenic to humans and causes severe central nervous system diseases. The third bacteriophage selected for the study was the uncharacterized phage I11m, specific for the clinical strain of E. coli D. We chose this phage because its host was often an isolated human pathogen. Besides the T4 phage, none of the phages was sequenced. Therefore the comparative analysis of their genomes was not possible. However, the diversity among E coli infecting phages is known to be highly wide-esprit. Comparisons of the T4-type genomes have shown that these phages share a typical ancestral genome but have been modified in numerous ways during their evolution. Generally, the phages that infect closely related host bacterial species are phylogenetically closer to each other than those that infect distantly related hosts. Therefore we can expect that the homology between coliphage proteins exists.
We have added the comment in the Discussion section.
Minor nomenclature error (E. coli is not in italic format)
It has been corrected
Round 2
Reviewer 2 Report
No added comments